# Heat Acclimation Does Not Protect Trained Males from Hyperthermia-Induced Impairments in Complex Task Performance

**DOI:** 10.3390/ijerph16050716

**Published:** 2019-02-28

**Authors:** Jacob F. Piil, Chris J. Mikkelsen, Nicklas Junge, Nathan B. Morris, Lars Nybo

**Affiliations:** Department of Nutrition, Exercise and Sports, Section for integrative physiology, University of Copenhagen, 2200 Copenhagen N, Denmark; jacob@nexs.ku.dk (C.J.M.); nicklas.junge@nexs.ku.dk (N.J.); morris@nexs.ku.dk (N.B.M.); nybo@nexs.ku.dk (L.N.)

**Keywords:** visuo-motor tracking, mathematics, motor performance, hyperthermia, core temperature, task complexity, heat stress

## Abstract

This study evaluated if adaptation to environmental heat stress can counteract the negative effects of hyperthermia on complex motor performance. Thirteen healthy, trained males completed 28 days of heat acclimation with 1 h daily exercise exposure to environmental heat (39.4 ± 0.3 °C and 27.0 ± 1.0% relative humidity). Following comprehensive familiarization, the participants completed motor-cognitive testing before acclimation, as well as after 14 and 28 days of training in the heat. On all three occasions, the participants were tested, at baseline (after ~15 min passive heat exposure) and following exercise-induced hyperthermia which provoked an increase in core temperature of 2.8 ± 0.1 °C (similar across days). Both cognitively dominated test scores and motor performance were maintained during passive heat exposure (no reduction or difference between day 0, 14, and 28 compared to cool conditions). In contrast, complex motor task performance was significantly reduced in hyperthermic conditions by 9.4 ± 3.4% at day 0; 15.1 ± 5.0% at day 14, and 13.0 ± 4.8% at day 28 (all *p* < 0.05 compared to baseline but not different across days). These results let us conclude that heat acclimation cannot protect trained males from being negatively affected by hyperthermia when they perform complex tasks relying on a combination of cognitive performance and motor function.

## 1. Introduction

Environmental heat stress has profound effects on human health and performance as the additional physiological strain or hyperthermia provoked by thermal stress has a negative impact on both cognitive functions [1,2,3,4,5,6,7] and physical exercise capacity [8,9,10,11,12]. This negative impact is a major societal challenge for public health, with recent meta-analyses demonstrating that occupational heat strain has implications for both health and productivity [13,14]. This heat strain may directly influence the billions of working people seasonally exposed to high heat scenarios, and indirectly influence individual income or national economies [15]. Furthermore, the derived effects on the ability to prevent poverty and secure socioeconomic stability is of importance for general health improvements [16,17]. Climate change implies that we are facing a future with expected increases in the intensity, frequency, and duration of heat waves [18] and in this context it is important to establish to what extent humans can adapt to high heat stress scenarios. It is well-known that heat acclimatization is associated with cardiovascular and sweating adaptations that may improve the capacity for heat dissipation and benefit the ability to perform prolonged strenuous activities in the heat [8,9,19]. However, many daily activities rely on fine motor skills and the ability to maintain attention (e.g., having to cognitively process visual and auditory inputs in order to elicit an appropriate motor response). It is not clear if adaptation to heat may protect cognitively dominated functions and fine motor performance from hyperthermia-induced impairments [20,21,22,23]. Brazaitis and Skurvydas (2010) reported that heat acclimation does not reduce the impact of hyperthermia on central fatigue during a maximal sustained contraction [24]. In contrast, Racinais and colleagues (2017) demonstrated that heat acclimation displayed a beneficial effect on the central nervous system i.e., improved executive functions and ability to maintain motor activation, whereas the peripheral nervous system, evaluated as electrically evoked M-wave and H-reflex amplitude, did not adapt to short-term heat acclimation [25]. 

Occupational field studies report significant reductions in productivity [26] and loss of effective work time [27] for workers adapted to high environmental heat. However, it is unclear if remaining effects on productivity and loss of effective work time is due to incomplete acclimatization or general limitations in human adaptability, and hence a limitation in the ability to maintain attention and motor function in high heat scenarios. We have recently developed a combined motor-cognitive test battery, which is able to detect hyperthermia-induced effects on complex motor performance [1] and to identify the influence of hypohydration on simple motor function and cognitively dominated tasks [28]. We applied this test battery before, during, and after a prolonged period of heat acclimation to explore if heat acclimatization induced by exercise-heat exposure would counteract the detrimental effects of hyperthermia on complex motor performance.

## 2. Methods

### 2.1. Participants 

Thirteen trained males (age: 40 ± 2 years, body mass: 79.5 ± 0.7 kg, height: 185.3 ± 1.0 cm, body fat percentage: 13.6 ± 1.1%, and maximal oxygen consumption (VO_2max_): 60 ± 1 mL O_2_·min^−1^·kg^−1^, (mean ± standard error of the mean (SEM)) participated in the study. Except for the below described heat acclimation procedure, the participants were instructed to maintain their regular training regimen throughout the intervention period. All participants received written and oral information on the experimental procedures and any risks and discomfort associated with the experiment before they provided written consent to participate in this study. Furthermore, this study was approved by the National Committee on Health Research Ethics (protocol number: 55907_v3_02012017). 

### 2.2. Experimental Design 

Overall design: Following a familiarization session in a cool room (described below), all participants completed a 28 day heat acclimation protocol, wherein all participants cycled five days a week, for 1 h, at 60% of the work load eliciting VO_2max_, in an environmental chamber at the University of Copenhagen. The chamber was maintained at 39.4 ± 0.3 °C and 26.6 ± 1.2% relative humidity. Motor-cognitive testing (described below) was carried out before (day 0), mid-way through (day 14), and at the end (day 28) of the heat acclimation protocol. All experimental testing and training was completed within the 28 days (including familiarization completed within 45 days). All participants were tested at the same time of day, at day 0, 14, and 28, to minimize the influence from circadian core temperature fluctuations. Participants were unaware of the researchers’ hypotheses and naive to the purpose of the study, but for obvious reasons, the participants were not blinded to the experimental conditions.

Familiarization session: Before initiating the experimental sessions, each participant was familiarized with all experimental procedures (e.g., cognitive testing and cycling in hot environment) to avoid significant learning effects during the actual testing sessions, which could potentially obscure the cognitive testing protocol. Similar to our previous studies [1,28], the participants completed a heating trial, consisting of a 60 min cycling bout (with optional electric fan use), in the laboratory at a room temperature of 38.5 ± 0.2 °C and 29.1 ± 2.1% relative humidity, to become familiar with heat exposure and the experimental procedures (described below). Furthermore, the familiarization protocol consisted of 1 day with 2 × 125 pre-trials of the complex motor task (1 h of training) separated by a short break (~5 min). Familiarization to the simple motor, simple cognitive, and the combined motor-cognitive test were limited to 10 pre-trials as pilot-testing demonstrated that participants had no increase in performance after ~5 pre-trials in the simple cognitive task and combined motor-cognitive tasks, and we did observe a carry-over effect from the complex motor task to the simple motor and combined motor-cognitive tasks. 

Cognitive testing sessions: During all cognitive testing sessions, all participants arrived at the laboratory 15 min before the start of the experiment. Participants then emptied their bladders into a sealed urine container, and individual urine samples was subsequently analyzed for urine specific gravity (USG) using a refractometer. All participants completed questionnaires reporting thermal comfort rating (TC) and temperature sensation rating (TS) and were weighed (without clothes), before a resting heart rate (HR) measure was obtained prior to entering the climatic chamber. Upon entering the chamber, the participants were seated for 5 min in an office chair and completed 20 further familiarization trials of the full protocol, prior to baseline testing, in order to avoid warm-up decrements in performance. Following completion of the baseline motor-cognitive testing, the participant transferred to their own bicycle mounted on a home-trainer (Tacx NEO smart, Tacx B.V., BW Wassenaar, The Netherlands) and exercised until reaching their maximal tolerable core temperature (T_rec_-hyperthermia (Hyper)) at day 0. This protocol was then repeated to ensure identical cut-off standards on day 14 and 28, where participants cycled until reaching the same T_rec_ as day 0. To prevent dehydration without influencing the participants core temperature, all subjects ingested warm (~37 °C) water matching their sweat loss (verified by weighing before and after commencing the motor-cognitive tests). The volume of water needed to maintain hydration levels, was estimated based on the familiarization heat test and adjusted for increased exercise time and sweat rate following acclimatization. Following exercise, the participants returned to the office chair and remained seated for 5 min, before initiating the motor-cognitive test battery. The familiarization session also included an incremental test to measure participants VO_2max_ and power output, which was used as the foundation for the work intensities in the subsequent training. During all trials and training, participants wore shorts, socks, and cycling shoes. 

During all motor and cognitive tests (specified below: see section Motor and cognitive performance testing), participants were seated on an office chair in front of a 24-inch computer monitor (Samsung, Seoul, Korea). The participant used their preferred (dominant) hand for all of the cognitive tests. The motor tasks involved recording pinch force via a strain gauge (Dacell, model UU3-K5, 5 Kgf, Cheongwon-gun, Korea), connected to a strain amplifier (Dacell, DN-AM-310, Cheongwon-gun, Korea) and the signal was subsequently digitalized and sampled at 500 Hz, with a data acquisition board, NI-USB-6008 (National instruments Inc., Austin, TX, USA). A customized script built on PYTHON (Python software foundation, Beaverton, OR, USA) was used for running the protocol. 

Acclimation status: In line with previous studies [8,19,29,30], resting T_rec_ and HR as well as exercise endurance in the heat, were assessed to verify acclimation status in participants. 

### 2.3. Measurements

Motor and cognitive performance testing: The motor-cognitive test protocol was adapted from Piil et al., 2017 ([1]; see method description of the multipart protocol for details). 

In brief, the protocol consisted of four different tasks: (1) Simple motor task (where a target force that the participant was required to pinch/adjust force to, with the aid of visual feedback, was displayed in the middle of the screen and the target number changed from one sequence to the next), (2) simple cognitive task (type the sum of four numbers), and (3) combined motor-cognitive (math addition task with the result provided via motor response/pinched force). For both cognitive/math tasks, four numbers (from 1 to 25) were displayed in each corner of the computer monitor. The participants were then required to calculate the sum of the four numbers (math addition) and either type the result (representing a simple cognitively dominated task) or pinch/adjust force via the strain gauge transducer, using thumb and index finger (representing a combined motor-cognitive task). The fourth task was a complex motor task with visuo-motor tracking (VMT). In this test, the participants were instructed to apply pinch force to a strain gauge transducer (with the thumb and index finger), thereby adjusting the vertical position of a cursor moving with constant speed across the monitor to, as accurately as possible, hit and stay within five target boxes. 

Each sequence of the test protocol started with the complex motor task, followed by a simple cognitive or combined motor-cognitive task, and finished with a simple motor task. The entire test consisted of 60 “sequences” i.e., 20 series of 3 task sequences (separated by a 3 s break between each sequence). Each of the above-mentioned tasks, lasted for 12 s and following each task, visual feedback (1 s) of the performance was provided with the score appearing in the bottom right corner of the screen and 2 s of transition time to the next task. The total sequence duration equaled 15 min. 

To characterize the reliability of the cognitive tests, the mean within-participant coefficient of variance (CV) as well as the intra-class coefficient of variance (ICC—both single and average were calculated for the baseline and hyper measures of all four tasks, as recommended by Atkinson and Nevill [31]: 

CV: Baseline and hyper CV were, 2.3% and 6.6% for the complex motor task, 6.6% and 7.4% for the combined motor-cognitive task, 0.9% and 3.5% for the simple motor task, and 1.1% and 2.8% for the simple cognitive task, respectively. 

ICC: Baseline and hyper ICC were, 0.900 and 0.883 (single), 0.964 and 0.958 (average) for the complex motor task, 0.660 and 0.573 (single), 0.853 and 0.801 (average) for the combined motor-cognitive task, 0.311 and 0.671 (single), 0.575 and 0.859 (average) for the simple motor task, and 0.639 and 0.138 (single), 0.841 and 0.324 (average) for the simple cognitive task, respectively.

Core temperature (T_rec_): was measured at rest (baseline) and end of exercise-heat exposure (hyper-denotes hyperthermia similar to our previous studies [1,28]) on all test days (day 0, day 14, and day 28) with a rectal thermometer (CTD85, Ellab, Copenhagen, Demark) inserted 7 to 10 cm beyond the anal sphincter.

Hydration: was assessed, by measuring USG before and after completing the motor-cognitive protocol, using a refractometer (ANTAGO, pocket refractometer, s/no P811580, Tokyo, Japan). USG was measured to ensure that all participants were hydrated (USG < 1.020 g·mL^−1^) before starting the trials and in combination with body weight assessment to track changes in hydration over time in the respective trials. During the trials, and to avoid (minimize) any dehydration and/or interference with the T_rec_, the participants were required to drink water equal to expected sweat loss as estimated from their familiarization trials. Body weight was assessed using a platform scale (InBody 270, InBody CO Ltd., Seoul, Korea).

Thermal comfort and sensation: Were assessed using questionnaires and VAS (visual-analog-scales) before (baseline) and after (hyper) the motor-cognitive test battery. The questionnaires were standardized (American Society of Heating, Refrigerating and Air Conditioning Engineers—ASHRAE standard 55), for ratings of thermal comfort (TC-scale from 0 (comfortable) to 3 (very uncomfortable) and temperature sensation (TS-scale from −3 (cold) to 3 (hot)). In addition, for a subgroup of 6 participants the VAS assessment of TC was applied in parallel to T_rec_ measures every 10 min during exercise to exhaustion trials in the heat to evaluate the relationship before and after acclimation. 

### 2.4. Statistical Analysis 

Changes in test scores and measurements across time and conditions were compared for simple motor, simple cognitive, combined motor-cognitive, and complex motor tasks and T_rec_, BM, USG, HR, TC, and TS were evaluated by a two-way repeated-measures analysis of variance test (2-way RM ANOVA) with the repeated factors of condition (2 levels: Baseline and hyper) and time (3 levels: day 0, day 14, and day 28). When a significant interaction or main effect in the 2-way RM ANOVA was found, post-hoc pairwise comparisons were performed with the Tukey correction test. The effect size of each *t*-test was calculated and reported as Cohen *d*, where 0.20 is a small effect size, 0.50 is a medium effect size, and 0.80 is a large effect size [32]. All statistical analyses were carried out in GraphPad Prism (version 7.02, GraphPad Software, La Jolla, CA, USA). Data are presented as mean ± SEM and the significance level was set at *p* = 0.05 (see Appendix A). 

## 3. Results 

### 3.1. Core Temperature (T_rec_)

Baseline T_rec_ values were lowered by an average of 0.4 ± 0.1 °C (*p* < 0.05) during the acclimation period. On each of the three test days (day 0, day 14, and day 28), T_rec_ increased on average by 2.8 ± 0.2 °C from baseline to the hyper time-point (all, *p* < 0.05) with no differences in T_rec_ for this condition across days (*p* < 0.05, see Table 1).

### 3.2. Hydration (BM and USG) 

Upon completion of the experimental trials, BM was slightly reduced by 0.69% (95% CI 0.47, 0.92%) compared to baseline (all *p* < 0.05) but did not differ between test days (*p* > 0.05, see Table 1) and maintained below 1% for all subjects. USG values remained below 1.020 and were similar across test conditions and days (*p* > 0.05, see Table 1).

### 3.3. Heart Rate (HR)

HR was higher in hyper compared to baseline on all days (*p* < 0.05). At test day 28, HR was lower at the hyper time-point compared to the hyper time-point at day 0 (*p* < 0.05). No other differences were observed for HR (see Table 1).

### 3.4. Ratings of Thermal Comfort (TC) and Temperature Sensation (TS)

TC and TS increased over time in all three test days with an average increase from low (neutral/zero for TC and approximately 1 for TS) at baseline to maximal levels (3.0 ± 0.0) for both scores at the hyper time-point (all *p* < 0.05—see Table 1). In a sub-group (*n* = 6) of the participants the VAS evaluation of thermal discomfort revealed that the relation between thermal discomfort and the core temperature was not changed by acclimation, but the slower progression of thermal discomfort over time and significantly prolonged exercise endurance time (day 0: 38.7 ± 2.4 min, day 14: 52.2 ± 1.1 min, and day 28: 64.3 ± 2.9; *p* < 0.05 between days) in the heat following day 14 and day 28 of acclimation related to an attenuated/slower rise in the subjects core temperature and TC (see Figure 1).

### 3.5. Combined Motor-Cognitive, Simple Cognitive, and Simple Motor Task Performances 

In the combined motor-cognitive and simple cognitive task the performance score was not affected by hyper (score on average 83.3 ± 1.7% (95% CI 77.7, 89.0%, *d* = 0.16) and 96.7 ± 0.6% (95% CI 94.7, 98.7%, *d* = 0.44), respectively) compared to baseline (score on average 85.0 ± 1.7% (95% CI 79.3, 90.6%) and 98.0 ± 0.3% (95% CI 96.9, 99.0%), respectively), and there were no significant differences between baseline and hyper across days (*p* > 0.05—day 0, day 14, and day 28). 

In the simple motor tasks, the performance score was lower in hyper compared to baseline (score on average 92.8 ± 1.2% (95% CI 90.3, 95.2%) and 96.9 ± 0.2% (95% CI 96.5, 97.3%), respectively—*p* = 0.05, *d* = 0.75), when all experimental days were considered.

### 3.6. Complex Motor Task Performance (VMT) 

In the complex motor task, baseline performance score was 71.6 ± 0.6% with no differences between days. However, at the hyper time-point, VMT performance was reduced by 9.4 ± 3.4% (95% CI 2.7, 16.0%, *d* = 0.83), 15.1 ± 5.0% (95% CI 5.4, 24.9%, *d* = 1.05) and 13.0 ± 4.8% (95% CI 3.6, 22.4%, *d* = 0.91) (all, *p* < 0.05) at day 0, day 14, and day 28, respectively, compared to baseline. No other differences in performance scores were observed (see Figure 2).

## 4. Discussion

In the present study, we evaluated simple and complex motor task performance as well as the ability to complete tasks relying on combinations of cognitive and motor function before, during, and after heat acclimatization. Of the applied tests, VMT was most sensitive to heat stress, while simple motor performance was slightly affected by hyperthermia and math calculation remained unaffected in the heat. Consequently, the ability to perform a complex motor task was markedly reduced when participants were exposed to uncompensable heat stress and neither medium (day 14) nor longer (day 28) heat acclimation seemed to be protective against the reduction in complex motor performance after exercise-induced heat stress, when the participants were exposed to uncompensable heat stress, which elevated their core temperatures, thermal comfort and sensation scores to maximal levels. During the intervention period, the participants displayed normal physical/physiological heat acclimation responses (i.e., lower resting core temperature and heart rate response at a given heat stress level, as well as longer exercise endurance during standardized heat stress exposure), but the present findings indicate that this could not protect against hyperthermia-induced impairments in complex motor performance. 

### 4.1. Contextualization to Occupational Settings

It is well established that occupational heat stress is associated with a significant increase in work related injuries and accidents [14,33,34], and during periods with high environmental heat strain (either shorter heat-waves or longer seasonal periods with high temperatures) the loss of productivity and effective labor time can become massive [13,27,35]. Indeed, these losses could relate to the observed detrimental effects of heat stress on complex motor performance, as this type of test, like many occupational tasks, rely on the ability to process visual, auditory, or tactile sensory information and provide an appropriate (skilled) motor response. Spread effects have previously been reported [1,2,6,20,28,36,37], but the effects of heat stress are particularly evident when the task complexity is high [1,2,4,6,20] and it appears to impact performance, hence productivity in real world occupations. 

It is well established that acclimatization can reduce heat strain and improve physical performance in the heat [29,38]. In conditions where the improved thermoregulatory capacity may prevent workers from hyperthermia, heat acclimatization may also benefit cognitively demanding tasks and tasks relying on fine motor skills. However, as demonstrated by the present study, heat acclimation cannot protect individuals from being affected by high levels of hyperthermia. 

Following twenty-eight heat acclimation sessions or approximately 1-month intervention, i.e., longer duration than previous supervised/controlled acclimatization studies, we still observe detrimental effects on the participants’ ability to perform complex motor task. Accordingly, observations have been reported in occupational settings where productivity remained affected in workers used to high heat exposure [26] and for military personal that have been heat acclimatized for 10 days [21]. In the latter study, significant physiological adaptations to heat matched the present study, however no positive adaptation in preventing cognitive performance decrements (i.e., the repeated acquisition task and time estimates were higher than during the participants’ first heat exposure) was observed [21]. In contrast, some studies indicate that heat acclimation may protect cognitive performance, both when core temperature is controlled to reach a fixed level [25] or not fixed [22] to the pre-acclimation level. These studies, utilized a singular task (i.e., rapid visual information processing, relying on sustained attention [22] and the one touch stocking test, relying on executive function [25]). In comparison, the present study utilized a verified multipart protocol, which included four different tasks (either relying primarily on cognitive function or a combination of motor performance and cognitive processing of visual inputs) alternating in a semi-randomized order [1], which also challenge attention and the ability to switch between tasks [39,40]. Therefore, the differences between observations in the previous studies compared with the present study, may relate to the applied test battery and/or the complexity of the involved tasks. Impaired VMT performance following exercise-induced hyperthermia seems to relate to thermal effects on central nervous system (CNS) function, while exercise or the prolonged endurance time on day 14 and 28 is expected to have no influence on the observed results or even opposite effects on VMT performance. Thus, aerobic exercise of similar intensity and duration (~ 1 h) has no negative effects on VMT performance when the rise in internal temperature is low or moderate [1]. Furthermore, meta-analyses by Lambourne and Tomprorowski 2010 [41] indicate that exercise has a small, but positive, effect on VMT and cognitively dominated tasks, and this is augmented if the intensity or duration is increased [42].

In addition, we observed that thermal comfort and thermal sensation remained similar when a given level of hyperthermia was reached (TS and TC compared at the same absolute core and skin temperature before and after heat acclimatization), which is in agreement with the observations from Racinais et al. [25], but in contrast to those reported previously by Brazaitis and Skurvydas [24]. However, in the latter study, T_rec_ was not fixed to the same level as before heat acclimation, again emphasizing that improved thermal comfort, perception or performance improvements with acclimatization relates to improved thermoregulatory capacity, rather than improved ability to tolerate high internal temperatures. Collectively, these findings indicate that heat acclimation does not alter thermal perception at a fixed core temperature, but may improve both CNS function and benefit thermal comfort when the improved capacity to dissipate heat allows for an attenuated disturbance of the internal temperatures—i.e., lower core and CNS temperatures. These findings outline the potential for and limitation of heat acclimatization to benefit workers’ performance and well-being. Considering a future where climate change implies more frequent exposure to highly stressful conditions, the present study supports the need for limiting the rise in environmental temperature as excessive thermal increases will lead to scenarios where humans. independent of adaptation/acclimatization ability, cannot avoid reaching hyperthermic levels and prevent detrimental effects on performance. 

### 4.2. Global and Public Health Perspectives 

Climate change implies that heat waves in the future will become more frequent, longer in duration, affect larger geographical areas of the world, and will likely negatively impact the global economy as well as workers’ health and productivity [13,16]. Therefore, it is necessary to identify solutions and strategies that may mitigate the negative effects [43]. Even though humans have a remarkable ability to adapt to environmental heat stress, the present findings emphasize the limitations for acclimatization to prevent the negative effects of high heat scenarios on productivity [27] and health [17]. 

The intervention period in the present investigation is amongst the longest scientific studies with supervised laboratory-controlled heat acclimatization. However, compared to occupational settings, where some workers may be exposed to seasonal heat stress for years, the adaptation period in the present investigation is obviously much shorter [21,22,24,25,29]. However, combined with observations from occupational field studies, reporting impaired productivity in experienced workers exposed to seasonal heat stress for many years, the present study signifies that the negative effects cannot be prevented in high heat stress scenarios. The complexity of occupational tasks may vary across industries, and considering that heat acclimation may protect simple tasks, it is possible that some functions, although less complex than the applied test in the present study, may be better protected in heat acclimatized workers [22,25]. However, occupations that require multitasking e.g., detection and integration of inputs from multiple sources followed by the selection and execution of an appropriate motor output, are likely vulnerable to hyperthermia.

## 5. Conclusions

The present study demonstrates that heat acclimation cannot protect trained males from being negatively affected by hyperthermia when they perform complex tasks relying on cognitive and motor performance under uncompensable heat stress scenarios. Combined with observations from occupational studies involving workers with many years’ experience and exposure to seasonal heat, the present findings signify that the negative effects of excessive environmental heat cannot be prevented or neglected. Heat acclimatization may indeed benefit physical performance, improve thermoregulatory capacity, and therefore improve thermal tolerance to a given heat load. However, in uncompensable heat stress conditions, the negative effects on complex motor tasks will remain. Considering climate change, these observations emphasize the importance of limiting the rise in environmental temperatures to levels that are within tolerable limits for working humans and prevent (or limit) the geographical and temporal spread of high heat scenarios.

## Figures and Tables

**Figure 1 ijerph-16-00716-f001:**
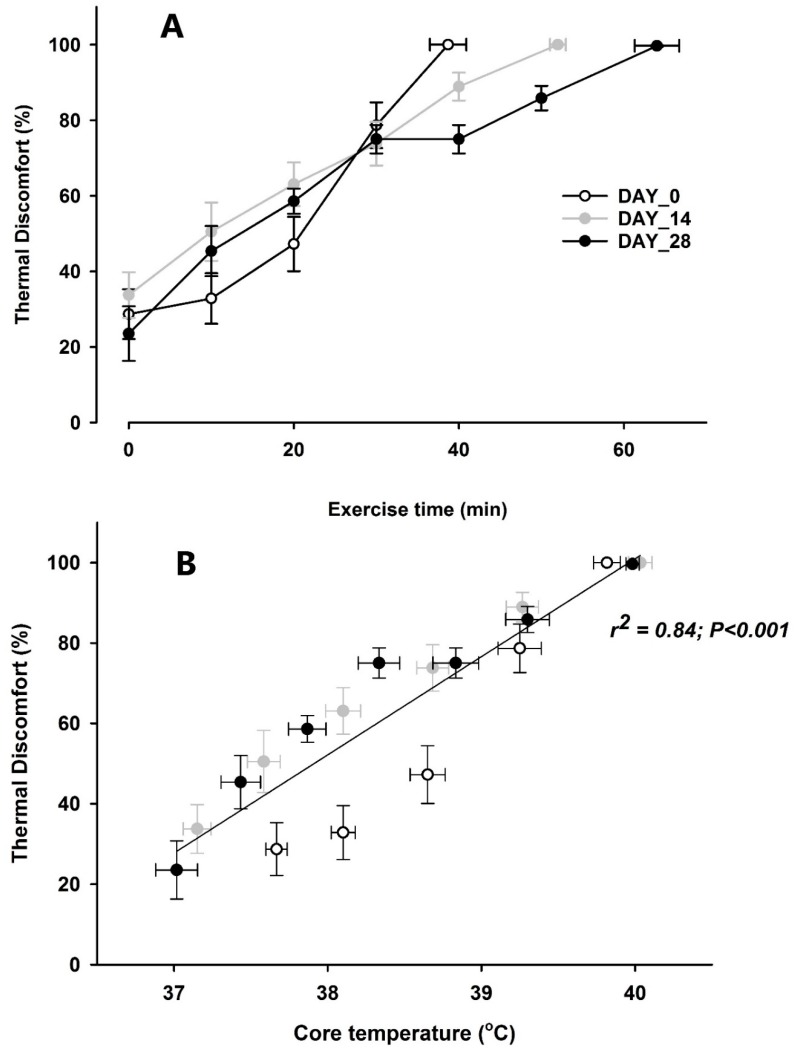
Thermal discomfort vs. exercise time and core temperature. (**A**) Thermal discomfort expressed as percentage of maximal thermal discomfort (i.e., 0% representing no thermal discomfort and 100% is maximal thermal discomfort) plotted against exercise time (min), at day 0 (white dots with black line), day 14 (gray dots with gray line), and day 28 (black dots with black line) and error-bars as SEM. (**B**) The correlation between thermal discomfort and core temperature, at day 0 (white dots), day 14 (gray dots), and day 28 (black dots) and error-bars (SEM) symbolizing the spread. The black line is the correlation line between parameters based on all observations.

**Figure 2 ijerph-16-00716-f002:**
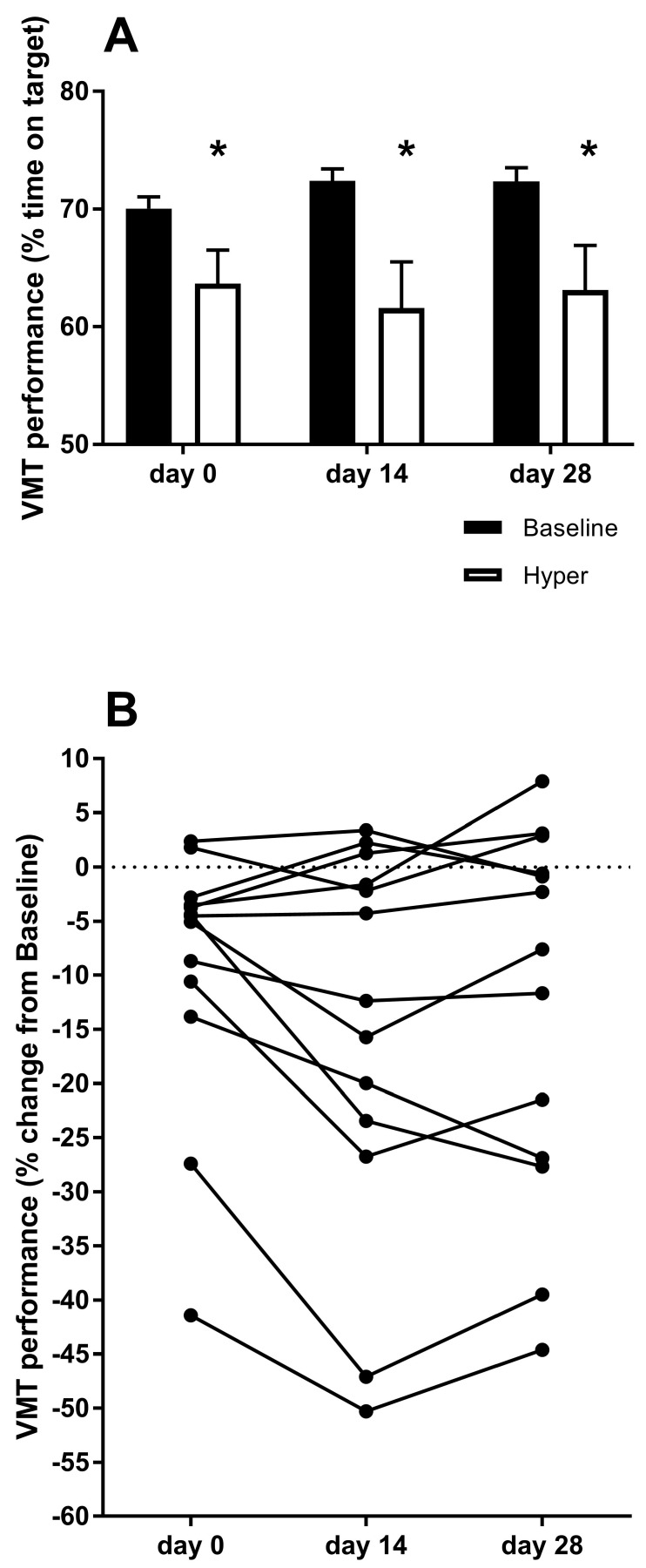
Complex motor task performance score. (**A**) Performance scores (expressed as the percentage of time on target) on day 0, day 14, and day 28 at baseline (black bars) and hyper (white bars) with error-bars representing SEM. (**B**) Individual changes (connected points representing each participant) in visuo-motor tracking (VMT) score from baseline to the hyper time-point on day 0 (before acclimation) and after day 14 and day 28 of acclimation. * significant different from Baseline, (*p* < 0.05).

**Table 1 ijerph-16-00716-t001:** Measures of physiological and psychological strain at baseline and hyperthermia—before (day 0), mid-way (day 14), and end (day 28).

	Day 0		Day 14		Day 28	
	Baseline	Hyper	Baseline	Hyper	Baseline	Hyper
T_rec_ (°C)	37.6 ± 0.1	40.1 ± 0.1 *	37.2 ± 0.1 ^#^	40.1 ± 0.1 *	37.1 ± 0.1 ^#^	40.1 ± 0.1 *
Body Mass (kg)	79.9 ± 1.8	79.4 ± 1.8 *	79.8 ± 1.8	79.3 ± 1.8 *	79.4 ± 1.8	78.6 ± 1.8 *
HR (bpm)	65 ± 3	131 ± 3 *	65 ± 3	126 ± 2 *	60 ± 3	120 ± 2 *^,^^
USG	1.016 ± 0.003	1.019 ± 0.003	1.011 ± 0.002	1.015 ± 0.002	1.016 ± 0.002	1.015 ± 0.002
TC	0.5 ± 0.0	3.0 ± 0.0 *	0.0 ± 0.0	3.0 ± 0.0 *	0.0 ± 0.0	3.0 ± 0.0 *
TS	1.5 ± 0.0	3.0 ± 0.0 *	1.0 ± 0.0	3.0 ± 0.0 *	1.0 ± 0.0	3.0 ± 0.0 *

Mean values across sessions and conditions; T_rec_ (rectal temperature in degree celsius), Body Mass (kilograms), HR (heart rate in beats per minute), USG (urine specific gravity), TC (thermal comfort), and TS (temperature sensation). Values are mean ± standard error of the mean (SEM). * significant different from baseline, ^#^ significant different from baseline (day 0), ^ significant different from hyper (day 0).

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
