# Peer review of "Heat Acclimation Does Not Protect Trained Males from Hyperthermia-Induced Impairments in Complex Task Performance"

_ijerph, 2019, doi:10.3390/ijerph16050716_

Round 1
Reviewer 1 Report
Comments to Author:
This research manuscript investigated the effects of heat stress on the cognitive performance of thirteen healthy males, completing a thermoregulatory adaptation period of 28 days. Subjects completed motor-cognitive testing, before thermoregulatory adaptation, and after 14 and 28 days of heat acclimation in baseline conditions following passive heating, and also following exercise induced hypothermia. The authors concluded acclimatisation to heat does not prevent cognitive performance impairment for complex tasks relying on both cognitive function and motor performance.
Abstract
Line 15, p.1. The abstract is very well written, although could be improved slightly with small grammatical changes that may improve the readability of the text for viewers. This may include minute changes such as the regular use of commas, to punctuate overly long sentences. For e.g., “acclimation(,) as well as after 14 and 28 days of heat acclimation.”
Introduction
The introduction is generally excellent and manages to include the most relevant and recent literature pertaining to the subject of heat stress and acclimation effects on cognitive performance. Although again the manuscript may be improved by small grammatical changes to enhance readability for the reader. For e.g., lines 31 (p.1), may be better presented as, “…additional physiological strain or hyperthermia provoked by thermal stressing has a negative impact”.
Also, for instance, lines 34-38 (p.1), could potentially be improved by halving this overly long sentence into two smaller succinct ones. Similarly, this process could be repeated for lines 5-9 (p.2). The authors structure the introduction quite well to highlight the gap in research that is being addressed by the manuscript, justifying the research requirement for the study.
Method
The methodology is very well described and presents a controlled within-subjects repeated measures design. Factors such as cognitive task familiarization procedures utilized (e.g., 2x125 pre-trials for the cognitive motor task), use of additional physiological markers such as USG to control for dehydration during heat exposure, show a strong scientific methodology, eliminating extraneous or uncontrolled factors, such as practice effects or existing dehydration. One area of suggestion, should it strengthen or add credibility to the conclusions drawn from the results, may be to calculate the statistical power of the study based on the number of participants used, in reference to the authors own previous research [references 1, 27 of the manuscript]. G*Power, provides free software for such purposes.
Line 24, p.2 – The authors may wish to present more simple abbreviations such as SEM or min in complete form, i.e., Standard error of the Mean and minutes, for first instance or use, as they have with all other technical terms. Although, these terms should be known to the reading audience, this could be a simple suggestion to improve readability for more general lay audiences. Note: The authors have presented SEM in complete as standard error of the mean on line 25, p. 4, please ctrl-search and replace first instance (i.e., Line 24, p.2)
Line 33, p. 2. Day0, day14, and day28, may be presented as more scientifically sound with a suitable change in label, for e.g., Day(0), or Day_28, or some variation thereof.
Line 4, p. 4. Is there a reason the exercise-heat exposure condition, “Hyperthermia”, is labelled, “Hyper”? If this has been used in previous research excellent, otherwise the authors may want to consider using the full-term, to provide clarity for the condition under observation, and simply for the ease of reading the statistical results section to follow. The reviewer understands this may create significant problems in the restructuring of tables etc. So a simple alternative, may be just to register the use of Hyper as “Hyperthermia”, by citing previous research, even the authors own studies [1, 27] if preferred.
Results
Results are generally very well-presented, using a tri-combination of tables, text and figures, with no discernible overlap, repeating, or excess information. Some areas of the presentation of results could be improved such as:
· Table 1 – The authors may wish to use alternative symbols to ‘$’, for a more scientifically sound reading manuscript. Symbols such as ‘^’, may suffice, or similar alternatives thereof.
· Table 1 – Abbreviations in brackets such as (kg) (bpm), and (°C), should include the unabbreviated form for the ease of the reader, e.g., kilograms, beats per minute and degrees Celsius.
· Figure 1 – Panel B. Mean and individual change scores between -5 and 10, are slightly overlapping as to be obfuscating. Three suggestions to improve the clarity of the figure for the reader would be export the image in a higher resolution format 300DPI or greater, or to slightly decrease the units on the y-axis from 5, or increase the length of the y-axis whilst maintaining the unit same ratio, i.e., 5.
Discussion
The findings reported by the authors are substantiated by the results presented, and highlight the overall implications of this important research in contrast to previous literature. The conclusions inferred by the results and contextualised in relation to the important areas of outcomes in terms of the implications of this manuscript for research in occupational settings, and global and health perspectives, demonstrates the breadth and importance of this study. The only major flaws of the discussion include the lack of discussion of limitations to the study and the proposal of future research directions. The authors approach the study limitations subject by acknowledging the length of their scientific protocol compared to occupational settings, is less extensive. One way to solve both issues simultaneously would be to acknowledge this limitation and offer a viable alternative for how future research may address this issue to create a new study. Other areas may also be improved by some minor adjustments, please see below.
Lines 10-11, p. 6. Where possible, when referring to conditions in the opening paragraph of the discussion, always remind and highlight for the reader the conditions tested, e.g., ‘heat stress’, is the ‘passive heat stress’ condition, and ‘hyperthermia’, is the ‘exercise-induced hyperthermia condition’.
Lines 1-2, p. 7. The reviewer posits using the terms ‘14 days’, and ‘28 days’, as opposed to day14 and day28 throughout the manuscript, may be preferable and improve the presentation of the authors’ work.
Lines 24-26, p. 7. Sentence structure and grammar may be improved by converting symbols into text, and shortening long sentences, especially in the discussion section, the use of symbols is more correctly used in the methodology section. For e.g., changing ‘~’, too approximately, and ‘/’ to and or. Even avoiding the over-use of abbreviations in-text such as, i.e., or using abbreviations only in brackets (i.e.,) improves sentence structure and grammatical flow for the reader, enhancing their understanding and the impact of your research on their comprehension of the research area, especially in the discussion and introduction sections where information can already tend to be quite technical.
Line 33, p. 7. Is there a better synonym for ‘clamp’, and ‘clamped’ that the authors could use?
Lines 16, p. 8. Revise. Insert, ‘in’, between “longest scientific”. Or re-arrange sentence structure to better convey meaning.
Lines 23-24, p. 8. The reviewer would urge the author to refrain from the use of placing whole sentences into brackets, in the discussion section. For e.g., “…is possible that some functions (less complex than the applied test in the present study) may be better …” Revise to: “…is possible that some functions, although less complex than the applied test in the present study, may be better …”
Similarly, revision of lines 25-26, to remove sentences from brackets would improve the readability and scientific soundness of the manuscript. The reviewer would like to thank the authors’ for their time and effort in the perusal of these comments, and commend them on their excellent work. Thank you.
Author Response
This research manuscript investigated the effects of heat stress on the cognitive performance of thirteen healthy males, completing a thermoregulatory adaptation period of 28 days. Subjects completed motor-cognitive testing, before thermoregulatory adaptation, and after 14 and 28 days of heat acclimation in baseline conditions following passive heating, and also following exercise induced hypothermia. The authors concluded acclimatisation to heat does not prevent cognitive performance impairment for complex tasks relying on both cognitive function and motor performance.
Answer: we would like to thank the reviewer for time spend and excellent comments that we have addressed to improve the quality of the paper – please find our point-to-point comments below.
Abstract
Line 15, p.1. The abstract is very well written, although could be improved slightly with small grammatical changes that may improve the readability of the text for viewers. This may include minute changes such as the regular use of commas, to punctuate overly long sentences. For e.g., “acclimation(,) as well as after 14 and 28 days of heat acclimation.”
Answer: Thanks for noticing, a coma has now been added after acclimation.
Introduction
The introduction is generally excellent and manages to include the most relevant and recent literature pertaining to the subject of heat stress and acclimation effects on cognitive performance. Although again the manuscript may be improved by small grammatical changes to enhance readability for the reader. For e.g., lines 31 (p.1), may be better presented as, “…additional physiological strain or hyperthermia provoked by thermal stressing has a negative impact”.
Answer: Thanks this has been corrected as suggested.
Also, for instance, lines 34-38 (p.1), could potentially be improved by halving this overly long sentence into two smaller succinct ones. Similarly, this process could be repeated for lines 5-9 (p.2). The authors structure the introduction quite well to highlight the gap in research that is being addressed by the manuscript, justifying the research requirement for the study.
Answer: We now address this as suggested “ This heat strain may directly influence the billions of working people seasonally exposed to high heat scenarios and indirectly influence individual income or national economies [15]. Furthermore, the derived effects on the ability to prevent poverty and secure the socio-economic stability is of importance for general health improvements [16,17]”
And
“In contrast, Racinais and colleagues (2017) demonstrated that heat acclimation displayed a beneficial effect on the central nervous system i.e. improved executive functions and ability to maintain motor activation, whereas the peripheral nervous system, evaluated as electrically evoked M-wave and H-reflex amplitude, did not adapt to short-term heat acclimation [25].”
Method
The methodology is very well described and presents a controlled within-subjects repeated measures design. Factors such as cognitive task familiarization procedures utilized (e.g., 2x125 pre-trials for the cognitive motor task), use of additional physiological markers such as USG to control for dehydration during heat exposure, show a strong scientific methodology, eliminating extraneous or uncontrolled factors, such as practice effects or existing dehydration. One area of suggestion, should it strengthen or add credibility to the conclusions drawn from the results, may be to calculate the statistical power of the study based on the number of participants used, in reference to the authors own previous research [references 1, 27 of the manuscript]. G*Power, provides free software for such purposes.
Line 24, p.2 – The authors may wish to present more simple abbreviations such as SEM or min in complete form, i.e., Standard error of the Mean and minutes, for first instance or use, as they have with all other technical terms. Although, these terms should be known to the reading audience, this could be a simple suggestion to improve readability for more general lay audiences. Note: The authors have presented SEM in complete as standard error of the mean on line 25, p. 4, please ctrl-search and replace first instance (i.e., Line 24, p.2)
Answer: The definition for SEM is provided as suggested - now stated in line 24 p.2 “[mean ± SEM (standard error of the mean)])”
Line 33, p. 2. Day0, day14, and day28, may be presented as more scientifically sound with a suitable change in label, for e.g., Day(0), or Day_28, or some variation thereof.
Answer: Good point we use the suggested Day_ throughout the manuscript. Changes from day0, day14 and day28 to Day_0, Day_14 and Day_28 – highlighted in yellow.
Line 4, p. 4. Is there a reason the exercise-heat exposure condition, “Hyperthermia”, is labelled, “Hyper”? If this has been used in previous research excellent, otherwise the authors may want to consider using the full-term, to provide clarity for the condition under observation, and simply for the ease of reading the statistical results section to follow. The reviewer understands this may create significant problems in the restructuring of tables etc. So a simple alternative, may be just to register the use of Hyper as “Hyperthermia”, by citing previous research, even the authors own studies [1, 27] if preferred.
Answer: this has now been addressed “Core temperature (Trec): was measured at rest (Baseline) and end of exercise-heat exposure (Hyper – denotes hyperthermia similar to our previous studies[1,28]) on all test days (Day_0, Day_14 and Day_28) with a rectal thermometer (Ellab Copenhagen, CTD85) inserted 7-10 cm beyond the anal sphincter.”
Results
Results are generally very well-presented, using a tri-combination of tables, text and figures, with no discernible overlap, repeating, or excess information. Some areas of the presentation of results could be improved such as:
· Table 1 – The authors may wish to use alternative symbols to ‘$’, for a more scientifically sound reading manuscript. Symbols such as ‘^’, may suffice, or similar alternatives thereof.
Answer: This has now been addressed and $ has been changed to ^.
· Table 1 – Abbreviations in brackets such as (kg) (bpm), and (°C), should include the unabbreviated form for the ease of the reader, e.g., kilograms, beats per minute and degrees Celsius.
Answer: This has now been added in the table text “Mean values across sessions and conditions; Tcore (rectal temperature, degree celcius), Body Mass (kilograms), HR (heart rate, beats per minute), USG (urine specific gravity), TC (thermal comfort) and TS (temperature sensation). Values are mean±SEM. * significant different from Baseline, # significant different from Baseline (Day_0), ^ significant different from Hyper (Day_0).”
· Figure 1 – Panel B. Mean and individual change scores between -5 and 10, are slightly overlapping as to be obfuscating. Three suggestions to improve the clarity of the figure for the reader would be export the image in a higher resolution format 300DPI or greater, or to slightly decrease the units on the y-axis from 5, or increase the length of the y-axis whilst maintaining the unit same ratio, i.e., 5.
Answer: The y-axis has now been lengthen and the resolution has now been increased to 600DPI. We have further removed the average bars, which make the graph more readable. This graph is now denoted Figure 2.
Discussion
The findings reported by the authors are substantiated by the results presented, and highlight the overall implications of this important research in contrast to previous literature. The conclusions inferred by the results and contextualised in relation to the important areas of outcomes in terms of the implications of this manuscript for research in occupational settings, and global and health perspectives, demonstrates the breadth and importance of this study. The only major flaws of the discussion include the lack of discussion of limitations to the study and the proposal of future research directions. The authors approach the study limitations subject by acknowledging the length of their scientific protocol compared to occupational settings, is less extensive. One way to solve both issues simultaneously would be to acknowledge this limitation and offer a viable alternative for how future research may address this issue to create a new study. Other areas may also be improved by some minor adjustments, please see below.
Lines 10-11, p. 6. Where possible, when referring to conditions in the opening paragraph of the discussion, always remind and highlight for the reader the conditions tested, e.g., ‘heat stress’, is the ‘passive heat stress’ condition, and ‘hyperthermia’, is the ‘exercise-induced hyperthermia condition’.
Answer: In accordance with the suggestion these lines has been changed to increase the readability
“Of the applied tests, the VMT was most sensitive to hyperthermia, while simple motor performance was slightly affected in the hyperthermic condition. Math calculation was now affected by exercise-induced hyperthermia”
Lines 1-2, p. 7. The reviewer posits using the terms ‘14 days’, and ‘28 days’, as opposed to day14 and day28 throughout the manuscript, may be preferable and improve the presentation of the authors’ work.
Answer: As previously mentioned we have changed day0, 14 and 28 to Day_0, 14 and 28 and 14 days and 28 days have been changes in accordance to Day_14 and Day_28.
Lines 24-26, p. 7. Sentence structure and grammar may be improved by converting symbols into text, and shortening long sentences, especially in the discussion section, the use of symbols is more correctly used in the methodology section. For e.g., changing ‘~’, too approximately, and ‘/’ to and or. Even avoiding the over-use of abbreviations in-text such as, i.e., or using abbreviations only in brackets (i.e.,) improves sentence structure and grammatical flow for the reader, enhancing their understanding and the impact of your research on their comprehension of the research area, especially in the discussion and introduction sections where information can already tend to be quite technical.
Answer: The abbreviations and symbols in this section and several places in the manuscript has now been changed, as suggested by reviewer. “Following twenty-eight heat acclimation sessions or approximately one month intervention (i.e. longer duration than previous supervised/controlled acclimatization studies)”
Line 33, p. 7. Is there a better synonym for ‘clamp’, and ‘clamped’ that the authors could use?
Answer: This has now been addressed and instead of clamp and clamped we use fixed (temperature).
Lines 16, p. 8. Revise. Insert, ‘in’, between “longest scientific”. Or re-arrange sentence structure to better convey meaning.
Answer: ‘In’ has been added as suggested.
Lines 23-24, p. 8. The reviewer would urge the author to refrain from the use of placing whole sentences into brackets, in the discussion section. For e.g., “…is possible that some functions (less complex than the applied test in the present study) may be better …” Revise to: “…is possible that some functions, although less complex than the applied test in the present study,may be better …”
Answer: This has been revised, as suggested by the reviewer.
Similarly, revision of lines 25-26, to remove sentences from brackets would improve the readability and scientific soundness of the manuscript. The reviewer would like to thank the authors’ for their time and effort in the perusal of these comments, and commend them on their excellent work. Thank you.
Answer: The bracket has now been removed as suggested.

Reviewer 2 Report
This study investigated the effect of long term heat acclimation (ie. > 12 exposures) on mitigating decrements in cognitive performance while experiencing heat stress in male participants. The results suggest 28 days of heat acclimation is not capable of mitigating cognitive performance when experiencing uncompensable heat stress.
General Comments:
- This paper is well written. The authors did a good job in clearly stating the purpose and developing a rational for its necessity and its application to society.
- A major caveat to the study was focusing exclusively on highly trained male participants. The conclusion as provided in the abstract, discussion section and conclusion section state heat acclimation cannot protect "humans" from being negatively affected by hyperthermia. This is conjecture, as the female response must also be examined to warrant such a statement.
- Furthermore, the mean aerobic power (i.e. VO2max = 60 mL-kg-min) of the sample is significantly above the average individual. This makes the conclusion difficult to extrapolate as the rate, and magnitude of acclimation would be different and the potential for decrements in cognitive performance may be different.
- The use of a single decimal place throughout the paper prevents the reader from gauging variation in the variables examined and the mean response observed. At times, there are numerical values without a decimal place.This is a major limitation when reading the current manuscript and for future replication. The use of one decimal place for values greater than 1.0 and two decimal places for values less than 1.0 is warranted to provide greater transparency and clarity.
Abstract:
- The conclusion stating humans cannot be protected from decrements in cognitive performance after heat acclimation requires revising. The current study examined a sample of 13 male participants. Such a statement cannot be made when only examining 13 participants and while focusing on the response of a specific sex. Please revise and make specific to the sex examined and acknowledge the need for further investigations to support the current findings before making general conclusions about the human response.
Introduction:
- A reference is warranted to strengthen the comment regarding current climate change and its impact on humans experiencing greater heat stress (Line: 38-40).
Methodology:
- I am curious to know the typical error of the cognitive assessments? Please provide a comment addressing the typical error so that the reader can ascertain its reliability. This would inherently strengthen/impact the results.
- A statement addressing the standardization towards the time of day heat acclimation was conducted and when cognitive performance assessments were performed is warranted. Variation in core temperature due to circadian rhythm is well supported as well as the intra-individual variation during day to day recordings.
- A statement addressing the level of physical activity of each participant throughout the study period is warranted. Were all subjects involved in exercise or sport? How can this impact cognitive performance? Current literature already supports the impact of physical activity on cognitive performance.
- The magnitude of physiological and performance adaptations to heat acclimation have shown to influenced by the period investigated within the calendar year. When was the current study conducted?
- Individuals who possess a high VO2max, as seen in the participants in this study, already display similar adaptations to heat acclimation, whereby the magnitude of change is limited. A statement and reference addressing this limitation is warranted especially when extrapolating its findings to the general public (ie.Humans).
Statistical Analysis:
- The current approach is suitable. However, the purpose of the study was to extrapolate its findings to impact all humans. Further analysis is warranted beyond an ANOVA to demonstrate its practical findings. The authors are strongly encouraged to include the Effect Size and its corresponding confidence interval (ES, 95% CI) to ascertain the magnitude of effect and its meaningful impact on society.
Results:
- The lack of decimal places prevents the reader from gauging any variation in the mean and changed scores. Please provide a single decimal for values >1.0 and at least two decimal places for values<1.0 throughout the manuscript and tables. As its stands, the numerical values lack transparency and prevent future meta-analysis from properly utilizing the findings presented.
- Please provide a statement in caption of Figure 1. "A" that describes what the error bars represent.
Discussion:
- This study exposed participants to uncompensable heat stress when examining cognitive performance. Statements towards heat acclimation failing to mitigate cognitive performance must be specific to the type of heat stress investigated. The authors are encouraged to rephrase the conclusion and specify the lack of evidence for HA to protect cognitive performance when experiencing uncompensable heat stress.
- I find it difficult to accept the authors conclusion when stating HA failed to mitigate cognitive performance tasks and its importance towards climate change. What was the length of each performance task? The impact of climate change on society's ability to work in the heat is beyond an assessment lasting a few minutes or an hour. To extrapolate the findings in the current study to the typical cognitive response to heat stress is highly speculative. The average work day is eight hours. Can an assessment lasting a few minutes provide a valid estimate we would expect in "humans" while exposed to heat stress over multiple hours during a work day due to climate change?
Author Response
General Comments:
- This paper is well written. The authors did a good job in clearly stating the purpose and developing a rational for its necessity and its application to society.
Answer: we would like to thank the reviewer for time spend and excellent comments that we have addressed to improve the quality of the paper – please find our point-to-point comments below.
- A major caveat to the study was focusing exclusively on highly trained male participants. The conclusion as provided in the abstract, discussion section and conclusion section state heat acclimation cannot protect "humans" from being negatively affected by hyperthermia. This is conjecture, as the female response must also be examined to warrant such a statement.
Answer: We have replaced humans with trained males. However, considering that trained males are those displaying the highest tolerance to heat stress (e.g. compared to untrained or aged males) and that the present findings/impact of hyperthermia in the unacclimatized state are similar to those observed in our previous studies (Piil et al. 2017 and 2018) with moderate trained/recreational active participants – it seems likely that the findings are of general importance for humans.
- Furthermore, the mean aerobic power (i.e. VO2max = 60 mL-kg-min) of the sample is significantly above the average individual. This makes the conclusion difficult to extrapolate as the rate, and magnitude of acclimation would be different and the potential for decrements in cognitive performance may be different.
Answer: We agree that these participants aerobic power is well above the average. However, trained endurance athlete are less vulnerable to heat stress than untrained or recreationally active individuals. Therefore, if prolonged heat acclimation do not protect endurance-trained participants; it is very likely that participants that are more vulnerable would also be affected. Therefore, we find it fair to extrapolate our finding to real-world occupations or leisure-time activities.
- The use of a single decimal place throughout the paper prevents the reader from gauging variation in the variables examined and the mean response observed. At times, there are numerical values without a decimal place.This is a major limitation when reading the current manuscript and for future replication. The use of one decimal place for values greater than 1.0 and two decimal places for values less than 1.0 is warranted to provide greater transparency and clarity.
Answer: As suggested we have now added decimals through the manuscript. However, for TC and TS the evaluation is in steps of 0.5 and therefore we have omitted it for those values – However, we report the VAS for TC where it was assessed with higher resolution.
Abstract:
- The conclusion stating humans cannot be protected from decrements in cognitive performance after heat acclimation requires revising. The current study examined a sample of 13 male participants. Such a statement cannot be made when only examining 13 participants and while focusing on the response of a specific sex. Please revise and make specific to the sex examined and acknowledge the need for further investigations to support the current findings before making general conclusions about the human response.
Answer: As suggested and included in both title and abstract, we have replaced humans with trained males. “These results let us conclude that heat acclimation cannot protect trained males from being negatively affected by hyperthermia when they perform complex tasks relying on a combination of cognitive function and motor performance.”
Introduction:
- A reference is warranted to strengthen the comment regarding current climate change and its impact on humans experiencing greater heat stress (Line: 38-40).
Answer: A reference has been added as suggested – Global risk of deadly heat, Mora et al. 2017
Methodology:
- I am curious to know the typical error of the cognitive assessments? Please provide a comment addressing the typical error so that the reader can ascertain its reliability. This would inherently strengthen/impact the results.
Answer: The mean within participant coefficient of variance (CV) and Inter-class coefficient of variance (single and avg.) at Baseline and at the Hyper time-point across the three test day has now been added.
“In order to characterize the reliability of the cognitive tests, the mean within-participant coefficient of variance (CV) as well as the intra-class coefficient of variance (ICC - single and avg.) were calculated for the Baseline and Hyper measures of all four tasks, as recommended by Atkinson and Nevill [31]:
CV: Baseline and Hyper CV were, 2.3% and 6.6% for the complex motor task, 6.6% and 7.4% for the combined motor-cognitive task, 0.9% and 3.5% for the simple motor task and 1.1% and 2.8% for the simple cognitive task, respectively.
ICC: Baseline and Hyper ICC were, 0.900 and 0.883 (single), 0.964 and 0.958 (avg.) for the complex motor task, 0.660 and 0.573 (single), 0.853 and 0.801 (avg.) for the combined motor-cognitive task, 0.311 and 0.671 (single), 0.575 and 0.859 (avg.) for the simple motor task, 0.639 and 0.138 (single), 0.841 and 0.324(avg.) for the simple cognitive task, respectively.“
- A statement addressing the standardization towards the time of day heat acclimation was conducted and when cognitive performance assessments were performed is warranted. Variation in core temperature due to circadian rhythm is well supported as well as the intra-individual variation during day to day recordings.
Answer: Thanks for noticing. We did in fact account for variations in core temperature by testing all subjects on the same time of day. “All participants was tested at the same time of day, at Day_0, 14 and 28, this to account for the core temperature fluctuations (i.e. circadian rhythm)”
- A statement addressing the level of physical activity of each participant throughout the study period is warranted. Were all subjects involved in exercise or sport? How can this impact cognitive performance? Current literature already supports the impact of physical activity on cognitive performance.
Answer: This has now been added “Participants were asked to maintain their regular training regimen throughout the intervention period.”
One of the reasons for inculding trained individuals, was to avoid the exercise effect on cognitive performance, as it has been observed that endurance exercise has a positive effect on cognition with a mean positive effect size of 0.2 for prolonged exercise (Mandolesi et al. 2018 and Lambourne and Tomprorowski 2010).
- The magnitude of physiological and performance adaptations to heat acclimation have shown to influenced by the period investigated within the calendar year. When was the current study conducted?
Answer: The study was carried out in the end of autumn to start spring. Therefore, the subjects was not exposed or exposed to any heat or hot conditions prior to or concomitant with the intervention.
- Individuals who possess a high VO2max, as seen in the participants in this study, already display similar adaptations to heat acclimation, whereby the magnitude of change is limited. A statement and reference addressing this limitation is warranted especially when extrapolating its findings to the general public (ie.Humans).
Answer: We have now added a figure (figure 1) that illustrate Thermal discomfort and time to exhaustion, as well as a correlation between thermal discomfort and core temperature. This show that even though individual with a high VO2max, can still markedly adapt to heat with prolonged exercise time. See also discussion above.
Statistical Analysis:
- The current approach is suitable. However, the purpose of the study was to extrapolate its findings to impact all humans. Further analysis is warranted beyond an ANOVA to demonstrate its practical findings. The authors are strongly encouraged to include the Effect Size and its corresponding confidence interval (ES, 95% CI) to ascertain the magnitude of effect and its meaningful impact on society.
Answer: Cohen d and 95% CI have been included for all tasks performance scores.
“Combined motor-cognitive, simple cognitive and simple motor task performances
In the combined motor-cognitive and simple cognitive task the performance score was not affected by Hyper (score on average 83.3±1.7% (95% CI 77.7, 89.0%, d=0.16) and 96.7±0.6% (95% CI 94.7, 98.7%, d=0.44), respectively) compared to Baseline (score on average 85.0±1.7% (95% CI 79.3, 90.6%) and 98.0±0.3% (95% CI 96.9, 99.0.%), respectively), and there were no significant differences between Baseline and Hyper across days (P>0.05 - Day_0, Day_14 and Day_28).
In the simple motor tasks, the performance score was lower in Hyper compared to Baseline (score on average 92.8±1.2% (95% CI 90.3, 95.2%) and 96.9±0.2% (95% CI 96.5, 97.3%), respectively – P=0.05, d=0.75), when all experimental days were considered.
Complex motor task performance (VMT)
In the complex motor task, Baseline performance score was 71.6±0.6% with no differences between days. However, at the Hyper time-point, VMT performance was reduced by 9.4±3.4% (95% CI 2.7, 16.0%, d=0.83), 15.1±5.0% (95% CI 5.4, 24.9%, d=1.05) and 13.0±4.8% (95% CI 3.6, 22.4%, d=0.91) (all, P<0.05) at Day_0, Day_14 and Day_28, respectively, compared to Baseline. No other differences in performance scores were observed (see figure 2).”
Results:
- The lack of decimal places prevents the reader from gauging any variation in the mean and changed scores. Please provide a single decimal for values >1.0 and at least two decimal places for values<1.0 throughout the manuscript and tables. As its stands, the numerical values lack transparency and prevent future meta-analysis from properly utilizing the findings presented.
Answer: This has now been addressed – in accordance with above mentioned procedure
- Please provide a statement in caption of Figure 1. "A" that describes what the error bars represent.
Answer: This has now been inserted into the figure caption “ Complex motor task performance score. (A) Absolute performance scores (expressed as the percentage (%) of time on target) with error-bars as SEM.”
Discussion:
- This study exposed participants to uncompensable heat stress when examining cognitive performance. Statements towards heat acclimation failing to mitigate cognitive performance must be specific to the type of heat stress investigated. The authors are encouraged to rephrase the conclusion and specify the lack of evidence for HA to protect cognitive performance when experiencing uncompensable heat stress.
Answer: We have now changed the first lines to capture that we observe no protective effects on complex motor performance on trained males, and that this is observed for uncompensable heat stress scenarios.
- I find it difficult to accept the authors conclusion when stating HA failed to mitigate cognitive performance tasks and its importance towards climate change. What was the length of each performance task? The impact of climate change on society's ability to work in the heat is beyond an assessment lasting a few minutes or an hour. To extrapolate the findings in the current study to the typical cognitive response to heat stress is highly speculative. The average work day is eight hours. Can an assessment lasting a few minutes provide a valid estimate we would expect in "humans" while exposed to heat stress over multiple hours during a work day due to climate change?
Answer: As stated in the method, each cognitive protocol were 15 min of length both at Baseline and at the Hyper time-point. We think it is fair to extrapolate our finding, because if the participants of the present study have decrements in performance scores, even though the present protocol only last 15 min. we would expect that workers, having to cognitive process visual, auditory and or tactile inputs and integrate this with an appropriate motor output, we would expect them to have lower performance.
Reviewer 3 Report
HA Does not protect humans from hyperthermia-induced impairments in complex motor performance. IJERPH 2019.
The study aimed to evaluated cognitive performance after exercise-heat stress and determine whether 14 or 28 days of heat acclimation (HA) protects against the detrimental effects of exercise-heat stress on complex cognitive processes. The authors should be commended for accomplishing a long-duration heat acclimation study. These types of studies are costly from both a time and financial stand point. Perhaps this is why no control group was included (?). Although there has been a recent boom in HA publications, there is still much we don’t know regarding the utility of HA. There are a number of critical, significant limitations to the current study design that must be addressed by the authors before publication can be considered. At minimum, authors must be forthright in acknowledging these limitations and aid the reader in interpreting the data in context of these limitations.
Major concerns:
The most significant short-coming of this study is the lack of control group consisting of exercise in temperate conditions. Without this group, one cannot parcel out the separate and combined effects of exercise and heat stress on cognitive performance. Studies exist demonstrating cognitive deficits with passive heat stress alone (Gaoua et al 2018) and with prolonged exercise (see the meta-analysis by Lambourne and Tomprorowski 2010). It is unknown whether it is the heat, exercise duration, or combination of the two that elicited the observed cognitive performance results. Particularly of value to those interested in HA would be days 14 and 28 where HA adaptations may or may not have been advantageous. The results must be interpreted with great caution and the authors need to be clear and contextualize the results within this limiting methodological decision.
In the case of this study, authors need to include more data to substantiate HA occurred and remained at day 14 and 28. Additional information regarding the intervention is also needed as it was unclear to this reviewer exactly how this intervention was employ. I would not be able to replicate this study with the current results. Below are specific concerns to be address:
Exercise duration for HA and for days 0, 14, and 28 need to be reported. It Trec was clamped and exercise intensity was set at 60% VO2max, then duration was variable. As mentioned previously, exercise duration plays a role in cognitive function. The authors must address this critical limitation and discuss how this may influence the observations.
P2L35: Could the authors please explain in more detail how the 28 days of HA was accomplished? Were there rest days? I understand they exercised in the heat Mon thru Fri then rest on the weekends, but were there additional rest days provided? Also, this accounts for 28 days yet all procedures were completed within 45 days. How do you account for the extra 17 days? Delaying testing after HA would have confounding results as it is well established that HA adaptations can decay rapidly, especially those of cardiovascular nature. Also, is it correct to state that subjects completed 20 days of exercise-heat stress? Or did they actually complete 14 and 28 days of exercise-heat stress. As you can see this is unclear and needs revision.
P3L16 Hydration status has an effect on cognitive function (Wittbrobt et al 2018 meta-analysis). BW and USG measures suggestion minimal dehydration<1.5% or so. However, it is not clear how fluid consumption occurred. What is meant by “encouraged to drink … to prevent dehydration without directly influencing Tcore”? Also, estimated fluid intake to sustain euhydration occurred at the baseline exercise-heat stress test (day 0). Did this level of fluid intake remain the same for all subsequent exercise-heat trials? A well-described HA adaptation is the increase in sweat rate necessitating additional fluid intake to offset sweat losses to sustain euhydration. Sweat rate and fluid intake data are missing and should be included so readers can evaluate this relationship.
P3L49: Were the order of cognitive tests randomized? If not, why and how could this have affected the results? Namely, the complex cognitive test occur last. How is it known that cognitive fatigue did not occur and was the reason for why complex cognitive performance was not sustained with HA?
Intro.
Well written.
Methods.
P2L22: It is interesting that subjects were males (only) and middle aged or excellent aerobic capacity. Could HA be protective in less fit populations where the influence of heat adaptation may be more striking?
P2L Please provide 1-2 sentences on how VO2max was achieved. Was it VO2max or peak?
P2L31: How / when was 60% VO2max confirmed?
P2L42 Please provide additional details on the use of a fan for wind speed. Direction? Speed? Air flow on which part of the body? Indirect wind flow? Etc.
Familiarization session: It appears that there is some type of learning/carry over effect with the cognitive test battery but authors familiarized subjects to reduce this potential confounding factor. Please report ICC’s to better demonstrate the elimination of the practice effect.
P3L13: Please use Trec for rectal temperature rather than Tcore. Trec is more descriptive of the actual measurement.
P3L20 So subjects did not wear a shirt?
P3L29 Sweat sodium content data not included nor was methodology either include this info or remove mention of this.
P3L29: There are many metrics to demonstration HA adaptation including the “classic” adaptations of lower exercise Trec, HR, and increased sweat rate. This reviewer is not convinced complete adaptation occurred given the current data included. It would be assumed that 14 and certainly 28 days of exercise-heat stress (with weekends off) would elicit HA but authors need to show this more convincingly. Perhaps include a comparison between day 0, 13, and 27 to show changes in exercise Trec, HR, and sweat rate (at least). Given that authors clamped Tec at 40 deg C at days 0 14 and 28, this cannot be used to establish HA; and a lower resting Trec is not convincing as this measure is subject to circadian rhythm. Was time of day standardized?
P4L15 Please provide written anchors (if any) for the thermal comfort and temp sensation scales. Also provide citations.
Results
Table 1. Pre- and post-exercise body mass was evaluated. Please report % body mass loss and include alterations in this calculation due to fluid intake, in possible or at least mention this wasn’t accounted for.
P5L22. So did HA actually worsen VMT performance? The trend is that VMT was lower… this seems counterintuitive given previous studies in this area.
How long after exercise-heat stress termination did the cognitive battery testing begin?
Discussion
Authors should set the stage for interpreting data. For example, HA seemed unprotective of VMT performance only after exercise-heat stress (of prolonged duration???) leading to a high Trec. What about exercise-heat stress of lower duration and lower Trec? How does this study’s ex duration and Tec related to previous work showing HA is beneficial in this regard?
P7L6 remove Na+ sodium comment or add data.
P7L44 Tskin was not measured (or data not included). Therefore, remove discussion about this measurement.
P7L49-51. Agreed, HA does not alter thermal perceptions at very high, fixed Trec like 40 deg C but may improve them at lower Trec. Part of the benefit of HA is the lower exercise Trec so there may indeed by a protective effect of HA, albeit at a lower Trec. However, the current study was not able to show this a exercise was terminated only when Trec – 40 deg C. This is a significant and important contextualization of the data. Had exercise duration be clamped along with intensity, Trec would certainly be lower due to HA and perhaps HA would then HAVE a protective effect as TC, TS, Tskin, and Trec would all likely be lower. This limitation must be discussed and authors should even consider changing the title to reflect this important difference. That is, HA may indeed have protective effects on cognitive after exercise-heat stress, just not after long (??) exercise or exercise resulting in high Trec. Also, it was exercise and heat stress used to induced hyperthermia.
Supplemental material (e.g., the database) lacked sufficient labelling of columns for full utility.
P9L18 and L21. Please add “seasonal” heat stress. The workers do not received continual heat stress throughout the year.
Conclusion
Authors need to contextualize their conclusions within the limitations of the study design. While I believe HA research is well positioned in the era of global warming, there seems to be some large leaps of logic connecting HA and global warming in the paper. As a fellow heat and hydration researcher, I understand the color of this type of writing, it is after all job security. However, tampering the language may be indicated here until additional data are available, particularly with the current data set and the glaring null effects of HA on protecting VMT performance.
Author Response
HA Does not protect humans from hyperthermia-induced impairments in complex motor performance. IJERPH 2019.
The study aimed to evaluated cognitive performance after exercise-heat stress and determine whether 14 or 28 days of heat acclimation (HA) protects against the detrimental effects of exercise-heat stress on complex cognitive processes. The authors should be commended for accomplishing a long-duration heat acclimation study. These types of studies are costly from both a time and financial stand point. Perhaps this is why no control group was included (?). Although there has been a recent boom in HA publications, there is still much we don’t know regarding the utility of HA. There are a number of critical, significant limitations to the current study design that must be addressed by the authors before publication can be considered. At minimum, authors must be forthright in acknowledging these limitations and aid the reader in interpreting the data in context of these limitations.
We would like to thank this reviewer for the constructive and comprehensive comments to our manuscript. Please find our point-to-point responses below as well as the revised paper in accordance with the input from all three reviewers.
Major concerns:
The most significant short-coming of this study is the lack of control group consisting of exercise in temperate conditions. Without this group, one cannot parcel out the separate and combined effects of exercise and heat stress on cognitive performance. Studies exist demonstrating cognitive deficits with passive heat stress alone (Gaoua et al 2018) and with prolonged exercise (see the meta-analysis by Lambourne and Tomprorowski 2010). It is unknown whether it is the heat, exercise duration, or combination of the two that elicited the observed cognitive performance results. Particularly of value to those interested in HA would be days 14 and 28 where HA adaptations may or may not have been advantageous. The results must be interpreted with great caution and the authors need to be clear and contextualize the results within this limiting methodological decision.
Answer: It is correct that we did not include a control group in the present study, and we fully acknowledge that studies including a control group is preferable in many/most situations and indeed that a control group would be required (of essential importance) if HA had improved the performance scores (i.e. counteracted the negative effect of hyperthermia that we in line with Gaoua et al 2018 observe for complex tasks, also in the unacclimatized state.).
However, in the present case a control group (with a similar variance as those observed for all our previous studies) would actually “add noise”/variance and increase the risk of stating that the HA group was not significantly different from control. Considering, that all scores and in particular VMT performance did not improve i.e. the negative effect of hyperthermia remained following HA seems to provide clear evidence that HA cannot prevent negative effects of hyperthermia. In relation to effects of exercise per se – we observe no negative effects in our previous studies when hyperthermia is prevented (see Piil et al. 2017) and previous studies report positive rather than negative effects of exercise on VMT performance (Roig et al 2012; Skriver et al 2014).
In the case of this study, authors need to include more data to substantiate HA occurred and remained at day 14 and 28. Additional information regarding the intervention is also needed as it was unclear to this reviewer exactly how this intervention was employ. I would not be able to replicate this study with the current results. Below are specific concerns to be address:
Exercise duration for HA and for days 0, 14, and 28 need to be reported. It Trec was clamped and exercise intensity was set at 60% VO2max, then duration was variable. As mentioned previously, exercise duration plays a role in cognitive function. The authors must address this critical limitation and discuss how this may influence the observations.
Answer: In accordance with the suggestion, we have included data on exercise endurance in the heat and illustrate (in new figure 1) how exercise time to exhaustion increases and thermal discomfort progress at slower rate during standardized heat exposure (demonstrating increased thermoregulatory capacity (however, not altering the relation between thermal discomfort and the rise in core temperature).
P2L35: Could the authors please explain in more detail how the 28 days of HA was accomplished? Were there rest days? I understand they exercised in the heat Mon thru Fri then rest on the weekends, but were there additional rest days provided? Also, this accounts for 28 days yet all procedures were completed within 45 days. How do you account for the extra 17 days? Delaying testing after HA would have confounding results as it is well established that HA adaptations can decay rapidly, especially those of cardiovascular nature. Also, is it correct to state that subjects completed 20 days of exercise-heat stress? Or did they actually complete 14 and 28 days of exercise-heat stress. As you can see this is unclear and needs revision.
Answer: We now clarify this and the reviewer is correct - the participant exercised every day from Monday to Friday and maintained normal training in the weekends. No additional rest days were provided.
This has now been clarified “Motor-cognitive testing (described below) was carried out before (Day_0), mid-way through (Day_14) and at the end (Day_28) of the heat acclimation protocol. All experimental testing and training was completed within the 28 days (incl. familiarization completed within 45 days). All participants was tested at the same time of day, at Day_0, 14 and 28, this to account for the core temperature fluctuations (i.e. circadian rhythm).”
This means that the 45 day included familiarization to all procedures, however, when participant initiated the experiment after familiarization all testing were completed within the 28 days.
Participants did complete 28 days of heat acclimations, therefore, we have denoted the tests Day14 and Day28.
P3L16 Hydration status has an effect on cognitive function (Wittbrobt et al 2018 meta-analysis). BW and USG measures suggestion minimal dehydration<1.5% or so. However, it is not clear how fluid consumption occurred. What is meant by “encouraged to drink … to prevent dehydration without directly influencing Tcore”? Also, estimated fluid intake to sustain euhydration occurred at the baseline exercise-heat stress test (day 0). Did this level of fluid intake remain the same for all subsequent exercise-heat trials? A well-described HA adaptation is the increase in sweat rate necessitating additional fluid intake to offset sweat losses to sustain euhydration. Sweat rate and fluid intake data are missing and should be included so readers can evaluate this relationship.
Answer: This is correct – and in accordance with Wittbrodt et al. 2018, we also recently report that dehydration (2% body mass deficit) aggravates performance (Piil et al. 2018) with approach utilizing similar tests as the present study. We therefore minimized any DH and have now added confidence intervals for the data on body weight changes -0.69 % ranging from CI 95% -0.47:-0.92% (with similar levels on days and below 1% BW change for all subjects)
The sentence has now been re-written to “To prevent dehydration without influencing the participants core temperature, all subjects ingested warm (~37°C) water matching their sweat loss (verified by weighing before and after commencing the motor-cognitive tests).
We did not measure the amount of water ingested during the sessions, only body mass before and after each session. Therefore, we cannot report sweat-rate.
P3L49: Were the order of cognitive tests randomized? If not, why and how could this have affected the results? Namely, the complex cognitive test occur last. How is it known that cognitive fatigue did not occur and was the reason for why complex cognitive performance was not sustained with HA?
Answer: The order was semi-randomized, so each sequences consisted of 3 tasks and all sequences started with a complex motor task, followed by either a combined motor-cognitive or a simple cognitive task and every sequence ended with a simple motor task (the order has been described in line 160-166 and for details we refer to Piil et al. 2017). This pattern continued for 15min. Therefore, as all tasks was conducted within the same battery of tasks and therefore we can concluded that HA did not have a protective effect on complex motor task performance.
Intro.
Well written.
Many thanks
Methods.
P2L22: It is interesting that subjects were males (only) and middle aged or excellent aerobic capacity. Could HA be protective in less fit populations where the influence of heat adaptation may be more striking
Answer: This is an interesting question. Previously we have observed similar decrement in performance in much less trained individual (Piil et al. 2017 and 2018). In the present study, we choose a group of trained participants to overcome the adaptations of training, as it would have been difficult to argue in case of a protective effect of HA had been observed, what had caused this increased training status or HA. We would expect that if well trained individuals would not be protected against hyperthermia-induced decrements, then less fit individuals also be affected – we do however now specifically mention the participants were trained males in the title as well as in the conclusions.
P2L Please provide 1-2 sentences on how VO2max was achieved. Was it VO2max or peak?
Answer: this has now been added “The familiarization session also included an incremental test to measure participants VO2max and power output, which was used to calculate workloads for the training sessions.”
P2L31: How / when was 60% VO2max confirmed?
Answer: We did not measure VO2max during the heat sessions – but during the pre-testing and participants then exercised at this intensity during all heat exposures (i.e. 60 % of the watts that elicited VO2max). This has now been added. “… at 60% of the work load eliciting VO2max”
P2L42 Please provide additional details on the use of a fan for wind speed. Direction? Speed? Air flow on which part of the body? Indirect wind flow? Etc.
Answer: No fanning were provided on test-days . i.e. fanning was only allow during the training sessions and not during days with standardized test. The wind speed was approximately 2m/s. The fan were placed within reach in front of the handlebars of the participant’s bikes.
Familiarization session: It appears that there is some type of learning/carry over effect with the cognitive test battery but authors familiarized subjects to reduce this potential confounding factor. Please report ICC’s to better demonstrate the elimination of the practice effect.
Answer: We have calculated the coefficient of variance (CV) and inter-class correlation with mean within participant at Baseline across the three test days.
“In order to characterize the reliability of the cognitive tests, the mean within-participant coefficient of variance (CV) as well as the intra-class coefficient of variance (ICC - single and avg.) were calculated for the Baseline and Hyper measures of all four tasks, as recommended by Atkinson and Nevill [31]:
CV: Baseline and Hyper CV were, 2.3% and 6.6% for the complex motor task, 6.6% and 7.4% for the combined motor-cognitive task, 0.9% and 3.5% for the simple motor task and 1.1% and 2.8% for the simple cognitive task, respectively.
ICC: Baseline and Hyper ICC were, 0.900 and 0.883 (single), 0.964 and 0.958 (avg.) for the complex motor task, 0.660 and 0.573 (single), 0.853 and 0.801 (avg.) for the combined motor-cognitive task, 0.311 and 0.671 (single), 0.575 and 0.859 (avg.) for the simple motor task, 0.639 and 0.138 (single), 0.841 and 0.324(avg.) for the simple cognitive task, respectively.“
P3L13: Please use Trec for rectal temperature rather than Tcore. Trec is more descriptive of the actual measurement.
Answer: This has now been changes throughout the manuscript.
P3L20 So subjects did not wear a shirt?
Answer: Correct participants did not wear a shirt during any of the sessions.
P3L29 Sweat sodium content data not included nor was methodology either include this info or remove mention of this.
Answer: This has now been removed throughout the manuscript.
P3L29: There are many metrics to demonstration HA adaptation including the “classic” adaptations of lower exercise Trec, HR, and increased sweat rate. This reviewer is not convinced complete adaptation occurred given the current data included. It would be assumed that 14 and certainly 28 days of exercise-heat stress (with weekends off) would elicit HA but authors need to show this more convincingly. Perhaps include a comparison between day 0, 13, and 27 to show changes in exercise Trec, HR, and sweat rate (at least). Given that authors clamped Tec at 40 deg C at days 0 14 and 28, this cannot be used to establish HA; and a lower resting Trec is not convincing as this measure is subject to circadian rhythm. Was time of day standardized?
Answer: We have no provided a new graph (denoted figure 1) that shows the correlation between exercise time and TC and Trec and TC.
P4L15 Please provide written anchors (if any) for the thermal comfort and temp sensation scales. Also provide citations.
Answer: The questionnaires was standardized (ASHRAE standard 55) as stated in the method. Start and end-point of both questioners has now been added to the description.
Results
Table 1. Pre- and post-exercise body mass was evaluated. Please report % body mass loss and include alterations in this calculation due to fluid intake, in possible or at least mention this wasn’t accounted for.
Answer: This has now been added with confidence intervals “Upon completion of the experimental trials, BM was slightly lowered by 0.69% (confidence 0.47:0.92%) compared to Baseline (all P<0< em="">.05) but did not differ between test days (P>0.05,”
P5L22. So did HA actually worsen VMT performance? The trend is that VMT was lower… this seems counterintuitive given previous studies in this area.
Answer: There is no trend. For 4 participants the VMT performance is slightly aggravated over time, however, 3 participant increase their performance and the rest of the participants have an unaltered performance score. The p-values between Baseline0 and 14 is P=0.339, between Baseline0 and 28 is P=0.363 and between Baseline14 and 28 P=0.999. More so, P=0.477 Day_0 vs Day_14 at the hyper time-point, as well as Day_0 vs Day_28 at the Hyper time-point P=0.996. Therefore, the slight decrease in performances, is dragged due to individual variations. To further increase the validity of the present findings 95% CI and effect size (Cohen d) has now been added to performance scores.
How long after exercise-heat stress termination did the cognitive battery testing begin?
Answer: As stated above, on the last day of heat acclimation, the final cognitive test was conducted on the participants 28th day of HA (Day_28).
Discussion
Authors should set the stage for interpreting data. For example, HA seemed unprotective of VMT performance only after exercise-heat stress (of prolonged duration???) leading to a high Trec. What about exercise-heat stress of lower duration and lower Trec? How does this study’s ex duration and Tec related to previous work showing HA is beneficial in this regard?
Answer:
P7L6 remove Na+ sodium comment or add data.
Answer: This has now been removed. We have added the effects on exercise tolerance in the heat instead.
P7L44 Tskin was not measured (or data not included). Therefore, remove discussion about this measurement.
Answer: Correct in these lines we do not discuss Tskin; but TS and TC (thermal sensation and comfort).
P7L49-51. Agreed, HA does not alter thermal perceptions at very high, fixed Trec like 40 deg C but may improve them at lower Trec. Part of the benefit of HA is the lower exercise Trec so there may indeed by a protective effect of HA, albeit at a lower Trec. However, the current study was not able to show this a exercise was terminated only when Trec – 40 deg C. This is a significant and important contextualization of the data. Had exercise duration be clamped along with intensity, Trec would certainly be lower due to HA and perhaps HA would then HAVE a protective effect as TC, TS, Tskin, and Trec would all likely be lower. This limitation must be discussed and authors should even consider changing the title to reflect this important difference. That is, HA may indeed have protective effects on cognitive after exercise-heat stress, just not after long (??) exercise or exercise resulting in high Trec. Also, it was exercise and heat stress used to induced hyperthermia.
Answer: We do agree that HA may protect cognitive performance if Trec would have been lower. However, the purpose of the present study was to see if HA would have a protective effect on motor-cognitive performance at a fixed Trec. As stated in the discussion section “It is well established that acclimatization can reduce heat strain and improve physical performance in the heat [38,39]. In conditions where the improved thermoregulatory capacity may prevent workers from hyperthermia, heat acclimatization may also benefit cognitively dominated tasks and tasks relying on fine motor skills. However, as demonstrated by the present study, heat acclimation cannot protect individuals from being affected by high levels of hyperthermia.”
Supplemental material (e.g., the database) lacked sufficient labelling of columns for full utility.
Answer: thanks for noticing, this has now been changed.
P9L18 and L21. Please add “seasonal” heat stress. The workers do not received continual heat stress throughout the year.
Answer: included as suggested.
Conclusion
Authors need to contextualize their conclusions within the limitations of the study design. While I believe HA research is well positioned in the era of global warming, there seems to be some large leaps of logic connecting HA and global warming in the paper. As a fellow heat and hydration researcher, I understand the color of this type of writing, it is after all job security. However, tampering the language may be indicated here until additional data are available, particularly with the current data set and the glaring null effects of HA on protecting VMT performance.
Answer: HA is identified as the most efficient way to mitigate negative effects of heat stress on physical performance and when effects of climate change are considered it is often assumed that humans on all aspects will adapt to new/higher levels of thermal stress. We therefore find it relevant to identify/establish if such effects can be expected for cognitively dominated tasks and combined cognitive-motor performances. This would be of importance for modelling of climate change effects across sectors and provide policy-relevant evidence for providing limits to heat-exposure in occupational setting or other scenarios where workers or populations are exposed to high heat stress
Round 2
Reviewer 2 Report
The authors have done a good job addressing the majority of concerns presented by the reviewers.
Author Response
Thanks to the reviewer for providing quality feedback.
Reviewer 3 Report
There was no author response to the following comment...
Authors should set the stage for interpreting data. For example, HA seemed unprotective of VMT performance only after exercise-heat stress (of prolonged duration???) leading to a high Trec. What about exercise-heat stress of lower duration and lower Trec? How does this study’s ex duration and Tec related to previous work showing HA is beneficial in this regard?
Answer:
Can the authors comment on whether exercise duration influenced VMT performance? The exercise time to target rectal temperature (40 deg. C) increased from Day 0 to 14 and 28 due heat acclimation enhancing heat dissipation mechanisms. Could exercise-induced fatigue have played a role in the observed no protective effect of HA on VMT? (see the meta-analysis by Lambourne and Tomprorowski 2010). Perhaps this is where the control group would have been advantageous?
This reviewer has no other suggestions/comments but would like to see these two comments addressed.
Author Response
Thanks for the additional comments – please see our responses below.
Authors should set the stage for interpreting data. For example, HA seemed unprotective of VMT performance only after exercise-heat stress (of prolonged duration???) leading to a high Trec. What about exercise-heat stress of lower duration and lower Trec? How does this study’s ex duration and Tec related to previous work showing HA is beneficial in this regard?
Answer: We apologize if this point was not explicitly addressed and emphasized in our previous answer. If the rise in core temperature is confined to “normal exercise levels” (~1degree above baseline) then we (Piil et al. 2017) and others (Roig et al 2012 and Skriver et al 2014) have previously demonstrated that VMT or other motor or cognitively dominated tasks remain unaffected - so it is only if subjects become hyperthermic or dehydration is superimposed that performance become affected. If heat acclimatization can prevent hyperthermia then an effect would be expected and in the discussion of the revised manuscript we address this aspect:
“It is well established that acclimatization can reduce heat strain and improve physical performance in the heat [38,39]. In conditions where the improved thermoregulatory capacity may prevent workers from hyperthermia, heat acclimatization may also benefit cognitively dominated tasks and tasks relying on fine motor skills. However, as demonstrated by the present study, heat acclimation cannot protect individuals from being affected by high levels of hyperthermia.”
Can the authors comment on whether exercise duration influenced VMT performance? The exercise time to target rectal temperature (40 deg. C) increased from Day 0 to 14 and 28 due heat acclimation enhancing heat dissipation mechanisms. Could exercise-induced fatigue have played a role in the observed no protective effect of HA on VMT? (see the meta-analysis by Lambourne and Tomprorowski 2010). Perhaps this is where the control group would have been advantageous?
Answer: As addressed above both we (Piil et al. 2017) and others observe no negative effects of easy/moderate or exhaustive exercise on subsequent VMT performance under conditions where the increase in the participants internal temperature remains low/moderate, whereas both exercise-induced and passively provoked hyperthermia impair subsequent VMT performance. Exercise per se or the duration is therefore unlikely the underlying mechanisms. In accordance Lambourne and Tomprorowski 2010, conclude that the mean effect size was 0.2 (95% CI 0.14-0.25) across all exercise modalities, i.e. moderate positive effect of aerobic exercise on VMT and cognitively dominated performances (and apparently with larger effects after intense and prolonged exercise (Smith et al. 2010).
One may argue that the lower VMT after exercise-induced hyperthermia (as well as passively induced) is a “fatigue response/sign” and in the discussion we now discuss this
See p9-10, line 50, 1-3 "Impaired VMT performance following exercise-induced hyperthermia seems to relate to thermal effects on CNS function, while exercise per se or the prolonged endurance time on Day_14 and 28 is expected to have no influence on the observed results or even opposite effects on VMT performance. Thus, aerobic exercise of similar intensity and duration (~ 1h) have no negative effects on VMT performance when the rise in internal temperature is low or moderate [1]. Furthermore, meta-analyses by Lambourne and Tomprorowski 2010 [42], indicate that exercise has a small, but positive effect on VMT and cognitively dominated tasks and this is augmented if the intensity or duration is increased [43]".